# The association between antihypertensive treatment and serious adverse events by age and frailty: A cohort study

James P. Sheppard[1]*, Constantinos Koshiaris[1], Richard Stevens[1], Sarah Lay-Flurrie[1], Amitava Banerjee[2], Brandon K. Bellows[3], Andrew Clegg[4], F. D. Richard Hobbs[1], Rupert A. Payne[5,6], Subhashisa Swain[1], Juliet A. Usher-Smith[7], Richard J. McManus[1]

1 Nuffield Department of Primary Care Health Sciences, University of Oxford, Oxford, United Kingdom, 2 Institute of Health Informatics, University College London, London, United Kingdom, 3 Columbia University Irving Medical Center, New York, New York, United States of America, 4 Academic Unit for Ageing & Stroke Research, University of Leeds, Leeds, United Kingdom, 5 Centre for Academic Primary Care, Population Health Sciences, University of Bristol, Bristol, United Kingdom, 6 Department of Health and Community Sciences, University of Exeter Medical School, Exeter, United Kingdom, 7 The Primary Care Unit, Department of Public Health and Primary Care, University of Cambridge, Cambridge, United Kingdom

* james.sheppard@phc.ox.ac.uk

**Data Availability Statement:** Data were obtained via a CPRD institutional licence. Requests for data sharing should be made directly to the CPRD (https://cprd.com). The Hospital Episode Statistics

## Abstract

### Background

Antihypertensives are effective at reducing the risk of cardiovascular disease, but limited data exist quantifying their association with serious adverse events, particularly in older people with frailty. This study aimed to examine this association using nationally representative electronic health record data.

### Methods and findings

This was a retrospective cohort study utilising linked data from 1,256 general practices across England held within the Clinical Practice Research Datalink between 1998 and 2018. Included patients were aged 40+ years, with a systolic blood pressure reading between 130 and 179 mm Hg, and not previously prescribed antihypertensive treatment. The main exposure was defined as a first prescription of antihypertensive treatment. The primary outcome was hospitalisation or death within 10 years from falls. Secondary outcomes were hypotension, syncope, fractures, acute kidney injury, electrolyte abnormalities, and primary care attendance with gout. The association between treatment and these serious adverse events was examined by Cox regression adjusted for propensity score. This propensity score was generated from a multivariable logistic regression model with patient characteristics, medical history and medication prescriptions as covariates, and new antihypertensive treatment as the outcome. Subgroup analyses were undertaken by age and frailty. Of 3,834,056 patients followed for a median of 7.1 years, 484,187 (12.6%) were prescribed new antihypertensive treatment in the 12 months before the index date (baseline). Antihypertensives were associated with an increased risk of hospitalisation or death from falls (adjusted hazard ratio [aHR] 1.23, 95% confidence interval (CI) 1.21 to 1.26), hypotension (aHR 1.32, 95% CI

data used in this analysis are re-used with permission from NHS Digital (https://digital.nhs.uk) who retain the copyright for that data. The Office for National Statistics provided mortality data. The Office for National Statistics and NHS Digital bear no responsibility for the analysis or interpretation of the data. Complete code lists used to define variables used in this analysis can be found at https://github.com/jamessheppard48/STRATIFY-BP/tree/Causal-inference-project.

**Funding:** This work received joint funding from the Wellcome Trust/Royal Society via a Sir Henry Dale Fellowship (ref: 211182/Z/18/Z; JPS) and the National Institute for Health Research (NIHR) School for Primary Care Research (SPCR; ref 418; JPS, RS, SLF, FDRH, RP, JAUS, RJM). RJMcM and FDRH acknowledge support from the NIHR Applied Research Collaboration (ARC) Oxford Thames Valley. RJMcM holds an NIHR Senior Investigator award. AB has received funding from NIHR, British Medical Association, UKRI and European Union. AC is supported by the NIHR Applied Research Collaboration Yorkshire & Humber (NIHR ARCYH) and Health Data Research UK, an initiative funded by UK Research and Innovation Councils, National Institute for Health Research and the UK devolved administrations, and leading medical research charities. BKB is supported by K01 HL140170 from the National Heart, Lung, and Blood Institute (NHLBI) (Bethesda, MD, USA). FDRH acknowledges part support from the NIHR Oxford University Hospitals Biomedical Research Centre. The funders had no role in study design, data collection and analysis, decision to publish, or preparation of the manuscript.

**Competing interests:** The authors have declared that no competing interests exist.

**Abbreviations:** aHR, adjusted hazard ratio; AUROC, area under the receiver operating characteristic curve; CI, confidence interval; CPRD, Clinical Practice Research Datalink; HES, hospital episode statistics; IMD, Index of Multiple Deprivation; IQR, interquartile range; NNH, needed to harm; NNT, number needed to treat; ONS, Office for National Statistics.

1.29 to 1.35), syncope (aHR 1.20, 95% CI 1.17 to 1.22), acute kidney injury (aHR 1.44, 95% CI 1.41 to 1.47), electrolyte abnormalities (aHR 1.45, 95% CI 1.43 to 1.48), and primary care attendance with gout (aHR 1.35, 95% CI 1.32 to 1.37). The absolute risk of serious adverse events with treatment was very low, with 6 fall events per 10,000 patients treated per year. In older patients (80 to 89 years) and those with severe frailty, this absolute risk was increased, with 61 and 84 fall events per 10,000 patients treated per year (respectively). Findings were consistent in sensitivity analyses using different approaches to address confounding and taking into account the competing risk of death. A strength of this analysis is that it provides evidence regarding the association between antihypertensive treatment and serious adverse events, in a population of patients more representative than those enrolled in previous randomised controlled trials. Although treatment effect estimates fell within the 95% CIs of those from such trials, these analyses were observational in nature and so bias from unmeasured confounding cannot be ruled out.

## Conclusions

Antihypertensive treatment was associated with serious adverse events. Overall, the absolute risk of this harm was low, with the exception of older patients and those with moderate to severe frailty, where the risks were similar to the likelihood of benefit from treatment. In these populations, physicians may want to consider alternative approaches to management of blood pressure and refrain from prescribing new treatment.

## Author summary

### Why was this study done?

- The benefits of blood pressure–lowering treatment have been widely studied, with recent reviews of the scientific literature suggesting increasing benefit as patients get older.

- The harms of blood pressure–lowering treatment are less well known, although another recent review of clinical trials showed that treatment is associated with acute kidney injury, hyperkalaemia (high blood potassium leading to medical complications), hypotension (low blood pressure) and syncope (fainting), but not falls or fracture.

- However, the trials included in these reviews are likely to have limited external validity, since participants are typically highly selected and diligently supported by trial teams in a way that does not reflect routine clinical practice.

- At present, there is little evidence to describe how the harms of antihypertensive treatment change as patients get older and develop frailty.

## What did the researchers do and find?

- This observational study utilised anonymised data from the electronic health records of patients in England. Those included were aged 40+ years, with high blood pressure, but had not previously been prescribed blood pressure–lowering treatment.

- A statistical analysis was undertaken to examine whether patients prescribed a blood pressure–lowering medication were more likely to experience a serious adverse event sooner, compared to those who were not prescribed such medications.

- In a total of 3,834,056 patients, blood pressure–lowering treatment was associated with an increased risk of hospitalisation or death from falls, hypotension, syncope (but not fracture), acute kidney injury, electrolyte abnormalities, and primary care consultations for gout.

- These risks were much higher in older patients and those with frailty. For example, in those aged 40 to 49 years, 3,501 patients would need to be treated for 5 years to cause a serious fall. However, for those aged 80 to 89 years, only 33 patients would need to be treated for the same period to cause a serious fall.

## What do these findings mean?

- Blood pressure–lowering treatment was found to be associated with an increased risk of serious adverse events.

- Across the whole population, the likelihood of experiencing this harm was very low.

- However, in older patients (aged 80+ years) and those with moderate to severe frailty, the risk of harm was notably increased.

- This analysis suggests that new prescription of blood pressure–lowering treatment in these older patients with frailty was just as likely to cause a serious fall, as it would prevent a stroke or heart attack.

## Introduction

An individual's risk of cardiovascular disease can be significantly reduced with antihypertensive treatment [1], and in recent years, hypertension management guidelines have recommended more intensive blood pressure–lowering strategies [2,3] based on trials demonstrating benefits in all age groups [4]. Antihypertensives are also among the most commonly prescribed medications in patients admitted to hospital with adverse drug reactions [5]. Consequently, guidelines [6,7] also recommend that physicians weigh the potential benefits of treatment against the potential harms when prescribing decisions are made [8]. However, such recommendations are hard to implement since little empirical evidence exists describing the association between antihypertensive therapy and serious adverse events.

A recent systematic review of 58 randomised controlled trials found some evidence that antihypertensive treatment is associated with acute kidney injury, hyperkalaemia, hypotension, and syncope, but not falls or fracture [9]. This review had some limitations, such as a

small sample size for certain outcomes and the selection bias associated with patients recruited to trials, compared to free-living people [10]. In addition, it was not possible to determine how treatment effects vary by patient-level characteristics such as age or frailty, due to the absence of individual patient data. Such information is critical for clinicians, particularly when making individualised treatment decisions in these subpopulations. Indeed, there is increasing support for the consideration of frailty status when prescribing of antihypertensive treatment [11], but in order to do this, better evidence is needed on the adverse effects of therapy in frailty subpopulations.

The present study, therefore, set out to accurately determine the association between antihypertensive treatment and subsequent serious adverse events, using nationally representative electronic health record data, by first replicating population treatment effects shown in meta-analyses of randomised controlled trials [9], and then studying how such serious adverse effects vary by age and frailty.

## Methods

The full methods for this study are described in the S1 Extended Methods. This study is reported as per the REporting of studies Conducted using Observational Routinely-collected Data (RECORD) guideline (S1 RECORD Checklist).

### Study design and setting

This was a retrospective observational cohort study, utilising electronic health record data from 2 datasets held within the Clinical Practice Research Datalink (CPRD); CPRD Gold, and CPRD Aurum (details of these datasets can be found the S1 Extended Methods). Both datasets have been shown to be representative of patients in England in terms of age, ethnicity, and deprivation [12,13]. These datasets were combined (excluding overlapping practices from the CPRD Aurum dataset) and linked at a patient level to Office for National Statistics (ONS) mortality data, basic inpatient hospital episode statistics (HES), and Index of Multiple Deprivation (IMD) data. The CPRD has global ethical approval for the use of anonymised electronic health records for research purposes, subject to approval of a study protocol by their Independent Scientific Advisory Committee. The protocol for this study was given prospective approval in February 2019 (ISAC protocol number 19_042) and is provided in the supporting information (S1 Protocol).

### Participants

Patients were eligible if they were aged 40 years or older, registered at a linked, "up-to-standard" general practice, had no previous prescription of antihypertensive therapy, and had records available after the study start date (1 January 1998). Eligible patients entered the cohort following their first systolic blood pressure reading ≥130 mm Hg (S2 Fig) [2,3]. Patients were excluded if they had no record of blood pressure measurement or a systolic blood pressure ≥180 mm Hg, since at this level treatment would be indicated regardless of risk of serious adverse events [2,3,6]. Exposure to antihypertensive medication was defined by the most recent prescriptions in the 12 months following cohort entry. The index date was defined at the end of this exposure period, after which patients were followed up for up to 10 years (S1 Fig).

Patient characteristics were determined from information recorded at any point prior to the index date. Patients exited the study on the study end date (31 December 2018), or when they transferred out of a registered CPRD practice, died, or experienced the specific outcome of interest.

## Outcomes

The primary outcome of this analysis was first hospitalisation or death with a primary diagnosis of a fall (defined according to ICD9 and ICD10 codes listed in S1 Table). Secondary outcomes were first hospitalisation or death with a primary diagnosis of hypotension, syncope, fractures, acute kidney injury, electrolyte abnormalities, and primary care attendance with gout (S1 Table and our GitHub page for codelists). In response to peer review comments, all-cause mortality and a composite outcome of serious adverse events were examined in further post hoc analyses. Serious adverse events were defined as first hospitalisation or death with a primary diagnosis of any of the conditions mentioned above (with the exception of gout, which was not included because it is typically less serious and usually only captured in primary care records).

## Exposure

The main exposure was prescription of any antihypertensive medication as defined in the British National Formulary (see S2 Table for details) [14]. Patients were allocated to the exposure group if they were prescribed at least 1 antihypertensive medication during the 12-month exposure window and medications at baseline were defined by the most recent prescriptions prior to the index date. Those not exposed during this period were included in the nonexposed group.

## Covariates

Predictors of antihypertensive treatment and the outcomes of interest were included as covariates in the analysis. These were selected based on clinical treatment guidelines [3], previous literature [15], and expert opinion and are detailed in the supporting information (S1 Extended Methods). Models were also adjusted for the database from which the data were derived (Gold or Aurum) and previous history of the outcome of interest.

## Sample size

A sample size of at least 88,380 patients (44,190 in each group) and 4,634 events was prespecified for analyses of each outcome of interest. This assumed a clinically significant increase in the rate of each adverse event with treatment of 10% [16], and an event rate of at least 0.5% per year in the nonexposed group, with 90% power and an alpha of 0.05. A conservative baseline event rate lower than previously reported in the literature [17,18] was chosen (2.2% to 7.7% per year in populations aged 55+ and 75+ years), due to the inclusion of younger patients than previously studied (40+ years).

## Propensity score estimation

Propensity scores were generated using multivariable logistic regression. Models included the covariates listed above, with continuous variables categorised to account for nonlinear associations with the outcome (the use of splines/fractional polynomials was explored but led to model convergence issues). Missing data were present for some covariates (see S1 Extended Methods for details), and these were dealt with using multiple imputation with chained equations (20 imputations) [19]. Propensity score model performance was assessed by the area under the receiver operating characteristic curve (AUROC) statistic, ratio of observed to expected probabilities (O/E ratio), and calibration plots. For propensity score matched analyses, treated patients were matched 1:1 to untreated patients, using the nearest

neighbour method (with calliper size restricted to 0.2), and standardised mean differences were estimated pre and post matching.

## Main analysis

For the primary analysis, propensity scores were included in Cox regression models along with previous history of the outcome of interest to examine the association between antihypertensive treatment and serious adverse events. For secondary analyses, (1) Cox regression models were adjusted for the same factors included in the propensity score models, with multiple imputation used to address missing data; (2) treatment effects were compared by Cox regression in patients matched by propensity score; and (3) inverse probability treatment weights were generated from the propensity score and used in a weighted Cox regression analysis with robust standard errors. The robustness of these methods was examined by comparing the results to published estimates from a meta-analysis of randomised controlled trials [9]. Model assumptions were checked through inspection of Schoenfeld residuals and survival curves for the main exposure. All analyses took an intention-to-treat approach and examined the time to event for a maximum of 10 years. Absolute risk differences were estimated (see S1 Extended Methods for details) and reported as the number of events per 10,000 patients treated per year, with confidence intervals (CIs) generated using bootstrap resampling (200 replications).

## Subgroup and sensitivity analyses

Analyses of treatment associations were examined in subgroups of the population by age (grouped into 10-year age bands) and frailty, determined using the electronic frailty index [20] and categorised into fit (score = 0 to 0.12), mild (score = >0.12 to 0.24), moderate (score = >0.24 to 0.36), and severe frailty (score = >0.36), using propensity score adjustment to control for confounding. Sensitivity analyses were undertaken to test the assumptions made to deal with missing smoking and deprivation data and examine the impact of competing risks on the treatment effect estimates, where the association between antihypertensives and falls (the primary outcome) was examined using a Fine-Gray competing risks model, with death from any cause (apart from falls) treated as a competing risk.

## Results

### Population characteristics

From a total population of 38,770,479 registered patients, 3,834,056 fulfilled the eligibility criteria (S2 Fig). The characteristics of individuals from both datasets were similar and are detailed in Table 1 and S3 Table. In the 12 months prior to the index date, 484,187 (12.6%) patients were prescribed antihypertensive therapy and included in the exposure group. Of these, 307,706 (63.6%) patients were prescribed 1 antihypertensive medication, 131,342 (27.1%) were prescribed 2, and 45,139 (9.3%) were prescribed 3 or more medications.

Patients entered the cohort throughout the period of observation (between 1998 and 2019; S3 Fig) and were followed up for a median of 7.0 years (interquartile range [IQR] 3.0 to 10.0 years). A total of 936,404 patients (28%) in the control group were prescribed an antihypertensive drug at some point during follow-up (S4 Table), but total treatment duration among these patients was significantly lower than in the exposure group (median 0.0 years [nonexposed; IQR 0 to 0.8 years] versus 6.0 years [exposed; IQR 2.0 to 10.0 years]).

The propensity score models included 31 covariates (S5 Table), displaying good discrimination (AUROC 0.82) and calibration (O/E ratio 1.36) predicting likelihood of treatment prior

**Table 1. Baseline characteristics of the study population.**

| Characteristic | No antihypertensive prescription during the 12-month exposure period (nonexposed) | | Antihypertensive prescription during the 12-month exposure period (exposed) | |
|---|---|---|---|---|
| | Mean/number | SD/% | Mean/number | SD/% |
| Total population | 3,349,869 | | 484,187 | |
| Age (years) (SD) | 55.9 | 12.1 | 61.7 | 12.9 |
| Sex (% female) | 1,666,304 | 49.7% | 245,498 | 50.7% |
| White ethnicity (%)* | 1,477,232 | 67.9% | 270,144 | 73.7% |
| Black ethnicity (%)* | 68,806 | 3.2% | 15,209 | 4.1% |
| South Asian ethnicity (%)* | 59,856 | 2.8% | 12,951 | 3.5% |
| Other ethnicity (%)* | 569,195 | 26.2% | 68,335 | 18.6% |
| High deprivation (IMD score of 5) (%)*† | 480,976 | 15.4% | 82,698 | 18.4% |
| Current smoking status (%)* | 770,592 | 24.4% | 98,215 | 21.4% |
| Alcohol consumption (heavy drinker) (%)* | 59,238 | 2.4% | 9,122 | 2.5% |
| Body mass index (kg/m$^2$) (SD) | 27.0 | 5.2 | 28.5 | 5.7 |
| Systolic blood pressure (mm Hg) (SD) | 141.4 | 10.8 | 150.8 | 13.7 |
| Diastolic blood pressure (mm Hg) (SD) | 83.2 | 9.0 | 88.2 | 11.8 |
| QRisk2 risk score (SD) | 10.8% | 11.0% | 22.2% | 15.5% |
| Electronic frailty index score (SD) | 0.04 | 0.05 | 0.08 | 0.06 |
| **Comorbidities** | | | | |
| Stroke (%) | 41,229 | 1.2% | 19,973 | 4.1% |
| Transient ischemic attack (%) | 19,309 | 0.6% | 9,404 | 1.9% |
| Myocardial infarction (%) | 18,560 | 0.6% | 25,893 | 5.3% |
| Heart failure (%) | 12,118 | 0.4% | 13,782 | 2.8% |
| Peripheral vascular disease (%) | 14,657 | 0.4% | 7,744 | 1.6% |
| Coronary artery bypass graft (%) | 3,404 | 0.1% | 6,170 | 1.3% |
| Angina (%) | 26,816 | 0.8% | 31,356 | 6.5% |
| Atrial fibrillation (%) | 34,551 | 1.0% | 24,218 | 5.0% |
| Diabetes (%) | 150,771 | 4.5% | 70,884 | 14.6% |
| Chronic kidney disease (%) | 28,219 | 0.8% | 21,309 | 4.4% |
| Cancer (%) | 116,589 | 3.5% | 24,303 | 5.0% |
| **Treatment prescriptions** | | | | |
| ACE inhibitors (%) | 0 | 0% | 187,209 | 38.7% |
| Angiotensin II receptor blockers (%) | 0 | 0% | 48,229 | 10.0% |
| Calcium channel blockers (%) | 0 | 0% | 141,454 | 29.2% |
| Thiazides and thiazide-like diuretics (%) | 0 | 0% | 148,652 | 30.7% |
| Beta-blockers (%) | 0 | 0% | 162,211 | 33.5% |
| Alpha-blockers (%) | 0 | 0% | 20,074 | 4.1% |
| Other antihypertensives (%)‡ | 0 | 0% | 8,143 | 1.7% |
| Statins (%) | 215,128 | 6.4% | 151,603 | 31.3% |
| Antiplatelets/anticoagulants (%) | 228,947 | 6.8% | 146,849 | 30.3% |
| Anticholinergics (%) | 295,101 | 8.8% | 40,000 | 8.3% |
| Antidepressants (%) | 604,344 | 18.0% | 88,625 | 18.3% |
| Hypnotics/anxiolytics (%) | 582,895 | 17.4% | 79,284 | 16.4% |
| Opioids (%) | 930,773 | 27.8% | 140,760 | 29.1% |

*Proportions based on the number of patients with information available (i.e., excluding those with missing values).

†IMD, indices of multiple deprivation; IMD score of 5 indicates patients in the highest quintile of deprivation (most deprived).

‡Other antihypertensives = centrally acting antihypertensives, direct renin inhibitors, and vasodilators.

to the index date (S4 Fig). A total of 429,800 treated patients were compared to 429,800 similar controls for the matched analysis (S6 Table).

## Primary outcome

During follow-up, a total of 63,561 patients (1.7%) were hospitalised or died following a fall, including 14,951 patients (3.1%) in the exposure group and 48,610 (1.5%) in the nonexposed group. In the primary analysis, using propensity score adjustment, antihypertensive treatment exposure was associated with an increased risk of hospitalisation or death from falls (adjusted hazard ratio [aHR] 1.23, 95% CI 1.21 to 1.26). This point estimate fell within the 95% CIs of estimates from meta-analyses of randomised controlled trials [9]. Analyses using multivariable adjustment, propensity score matching, and inverse probability treatment weighting produced similar results (Fig 1). The overall absolute risk of falls with antihypertensive treatment was very low, with just 6 events (95% CI 6 to 7) per 10,000 patients treated per year, equivalent to a number needed to harm (NNH) of 431 and 158 over 5 and 10 years, respectively (Table 2).

## Secondary outcomes

Antihypertensive treatment exposure was also associated with an increased risk of hospitalisation or death from hypotension, syncope, acute kidney injury, electrolyte abnormalities, and primary care consultations for gout, but not fracture (Fig 1 and Table 2). Once again, the point estimates for each outcome were similar across analytical approaches and fell within the 95% CIs of estimates from meta-analyses of randomised controlled trials [9], with the exception of hypotension and acute kidney injury (Fig 1). The absolute risk of serious adverse events with antihypertensive treatment was low for each individual outcome (Table 2). Post hoc analyses confirmed that antihypertensive treatment is associated with an increased risk of any adverse event (examined as a composite outcome) but overall a reduced risk of all-cause mortality (S7 Table).

## Subgroup and sensitivity analyses

The relative association between antihypertensive treatment and serious adverse events increased with age for falls, acute kidney injury, electrolyte abnormalities, and gout (Fig 2). These trends were less obvious in subgroups by baseline frailty (Fig 3). However, the estimated absolute risk of serious adverse events did increase substantially with both age and frailty, particularly for falls, acute kidney injury, and electrolyte abnormalities (Figs 2 and 3). For example, the absolute risk of hospitalisation or death from a fall with antihypertensive treatment in individuals aged 40 to 49 years was 1 event (95% CI 0 to 1) per 10,000 patients treated per year (equivalent to a number needed to harm [NNH] of 3,501 at 5 years and 1,751 at 10 years). In those aged 80 to 89 years, this was increased to 61 events (95% CI 52 to 70) per 10,000 patients treated per year (equivalent to an NNH of 33 and 16 at 5 and 10 years, respectively). Similarly, in fit patients, the absolute risk of a serious fall with antihypertensive treatment was 5 events (95% CI 4 to 5) per 10,000 patients treated per year (equivalent to an NNH of 433 at 5 years and 217 at 10 years). However, in those with severe frailty, this was increased to 84 events (95% CI 29 to 141) per 10,000 patients treated per year (equivalent to an NNH of 24 and 12 at 5 and 10 years, respectively).

Sensitivity analyses, examining different ways of dealing with missing smoking and IMD data, produced similar results to the primary analysis (S8 Table). Further sensitivity analyses, using a competing risks approach to examine the primary outcome, also found no difference between the sub-hazard ratio for serious falls (adjusted sub-hazard ratio 1.27, 95% CI 1.24 to 1.30) and the aHR from the primary analysis (S8 Table).

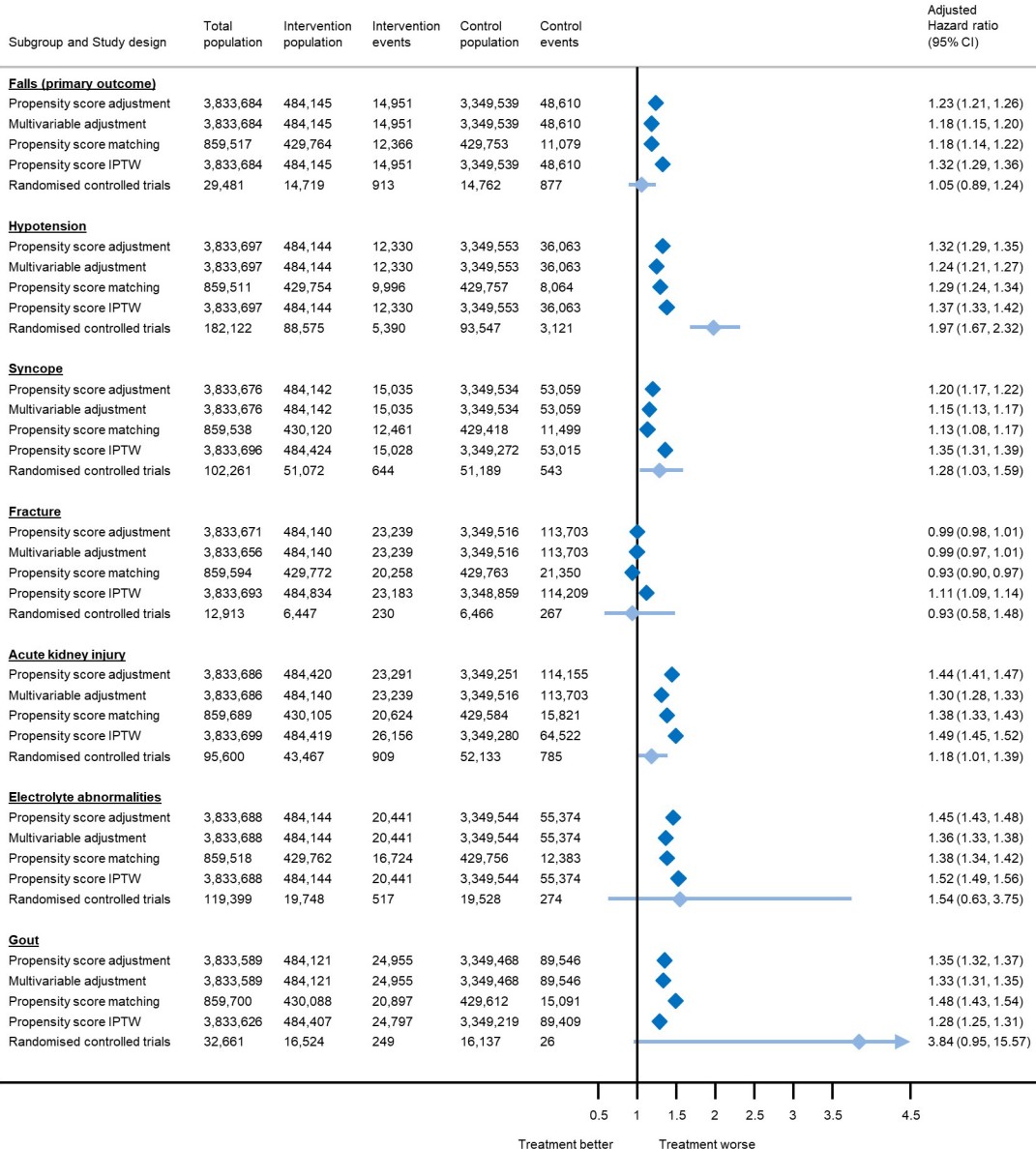

**Fig 1. Association between antihypertensive treatment and serious adverse events leading to hospitalisation or death, based on analyses of electronic health records and meta-analyses of randomised controlled trials.** Estimates from randomised controlled trials were derived from a previously published meta-analysis [9], and represent risk ratios rather than hazard ratios. For rare events such as the outcomes presented here, these would be expected to be equivalent. The total number of patients included in each analysis varies due exclusion of participants who experienced the outcome of interest on the index date, model convergence, and variation in the matching algorithm. CI, confidence interval; IPTW, inverse probability treatment weights.

## Discussion

### Summary of main findings

In this observational study of 3.8 million patients, previously untreated and with raised systolic blood pressure, antihypertensive treatment was associated with an increased risk over the subsequent decade of hospitalisation or death from falls, hypotension, syncope (but not fracture), acute kidney injury, electrolyte abnormalities, and primary care consultations for gout.

**Table 2. Hazard ratios, absolute risk differences, and numbers needed to harm to cause 1 outcome at 5 and 10 years.**

| Outcome | Unadjusted analyses | | Adjusted analyses* | | Absolute risk difference (additional events per 10,000 patients per year) | | Number needed to harm | |
|---|---|---|---|---|---|---|---|---|
| | Hazard ratio | 95% CI | aHR | 95% CI | Events | 95% CI | 5 years | 10 years |
| Falls (primary outcome) | 2.19 | 2.15 to 2.23 | 1.23 | 1.21 to 1.26 | 6 | 6 to 7 | 431 | 158 |
| Hypotension | 2.43 | 2.39 to 2.48 | 1.32 | 1.29 to 1.35 | 7 | 6 to 7 | 434 | 153 |
| Syncope | 2.02 | 1.98 to 2.05 | 1.20 | 1.17 to 1.22 | 5 | 5 to 6 | 429 | 183 |
| Fracture** | 1.45 | 1.43 to 1.47 | 0.99 | 0.97 to 1.01 | 0 | −1 to 0 | - | - |
| Acute kidney injury | 2.92 | 2.88 to 2.96 | 1.44 | 1.41 to 1.47 | 16 | 15 to 17 | 174 | 64 |
| Electrolyte abnormalities | 2.64 | 2.60 to 2.68 | 1.45 | 1.43 to 1.48 | 14 | 14 to 15 | 205 | 72 |
| Gout | 1.99 | 1.97 to 2.02 | 1.35 | 1.32 to 1.37 | 13 | 12 to 14 | 135 | 79 |

*Models adjusted for propensity score.

**Absolute risk difference too small to estimate number needed to harm

aHR, adjusted hazard ratio; CI, confidence interval.

Overall, serious adverse events were rare and the absolute risk of harm from treatment was very low. However, in older patients (aged 80+ years) and those with moderate to severe frailty, the absolute risk of harm was notably increased.

These data confirm the association between antihypertensive treatment and serious adverse events [9] and, to our knowledge, show for the first time how an individual's absolute risk of harm changes with increasing age and frailty. In older patients, the absolute risk of harm from a fall with treatment ($NNH_5 = 33$) was found to be very similar to the likelihood of benefit (number needed to treat [$NNT_5$] = 38) [21], and in such situations, the decision about whether or not to prescribe treatment is more finely balanced. With this in mind, recent calls to remove age-related blood pressure treatment thresholds from international guidelines [21,22] should be considered with caution. These findings can also be used by clinicians to guide individualised treatment decisions in partnership with patients. While patient choice remains key for all treatment decisions, the combination of advanced old age and increasing frailty severity may be a particular situation in which the balance of risk tips in favour of a more circumspect approach to treatment.

## Comparison with previous literature

Very few previous studies have attempted to quantify the relationship between antihypertensive treatment and serious adverse events and how these change with increasing age and frailty. A previous systematic review found evidence that antihypertensive treatment is associated with acute kidney injury, hyperkalaemia, hypotension, and syncope, but not falls or fracture [9]. This was also reported in another review focussing exclusively on falls [23], although both studies included heterogeneous populations and may have underrepresented patients with advancing age and frailty [24]. Some trials focussing on older populations have presented results stratified by frailty and concluded little association between treatment and serious adverse events [25,26]. However, these studies typically recruited healthier older people and were therefore not representative of those with moderate to severe frailty living in the community [27].

Previous observational studies have produced inconclusive findings when examining the association between antihypertensive treatment and adverse events, with some showing an association with falls and others not [15,16,23,28]. The present analysis examined a large

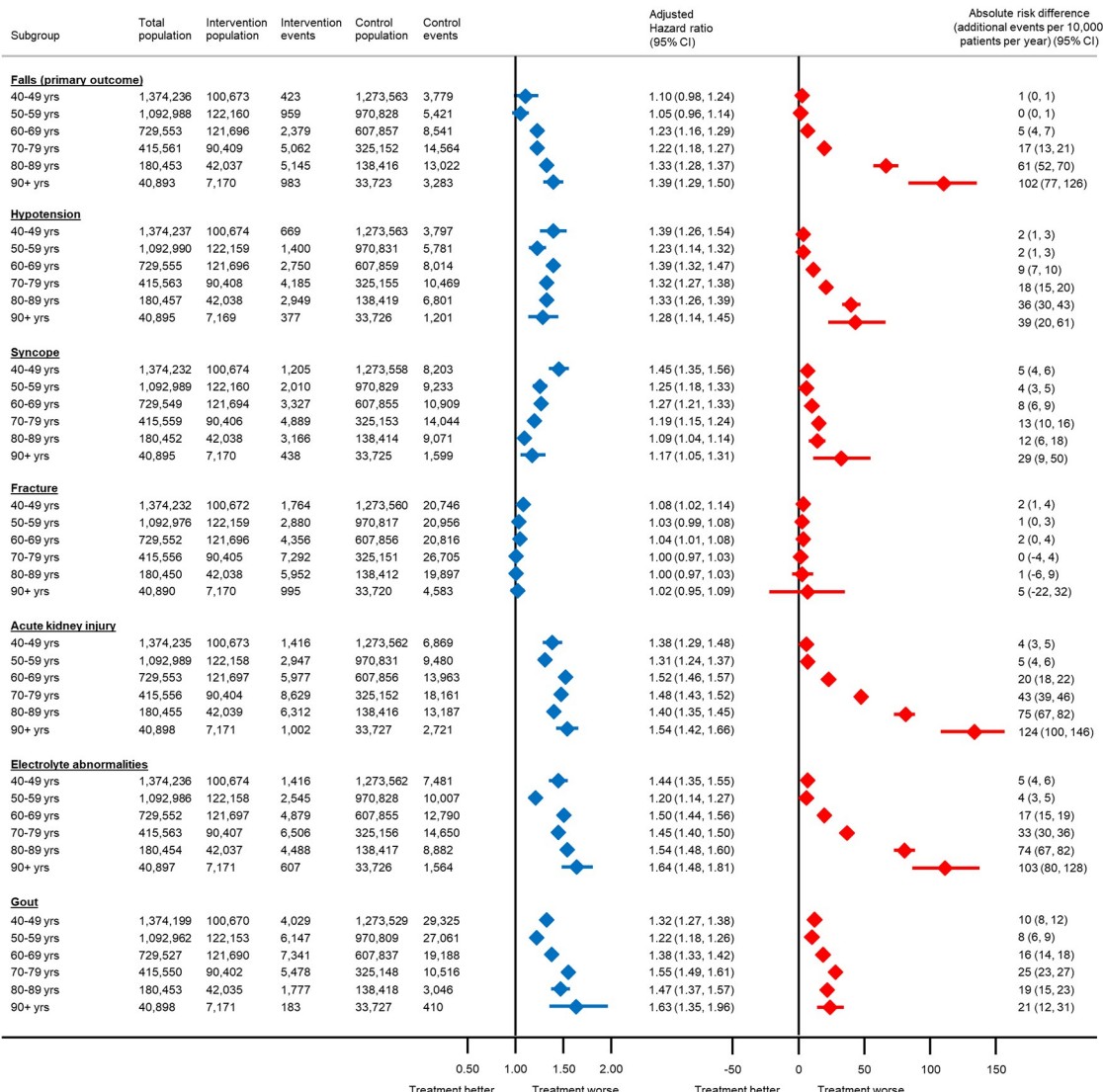

**Fig 2. Association between antihypertensive treatment and serious adverse events leading to hospitalisation or death, by age at the index date.** The total number of patients included in each analysis varies due to the exclusion of participants who experienced the outcome of interest on the index date. Models adjusted for propensity score. CI, confidence interval.

population, more generalizable than those included in previous trials, including a significant proportion of patients at older age and with moderate to severe frailty [24]. Not only was antihypertensive treatment found to be associated with falls, but the absolute risk increase with treatment was shown to be notably higher in the populations underrepresented in previous trials [24]. The lack of association between antihypertensive treatment and fracture has been reported previously [29] and may be partly explained by that fact that some fractures are not caused by syncope and falls and, therefore, may not be directly related to antihypertensive treatment.

## Strengths and limitations

To our knowledge, this was the largest population-based analysis of serious adverse events associated with antihypertensive therapy conducted to date [15], and the first to examine how

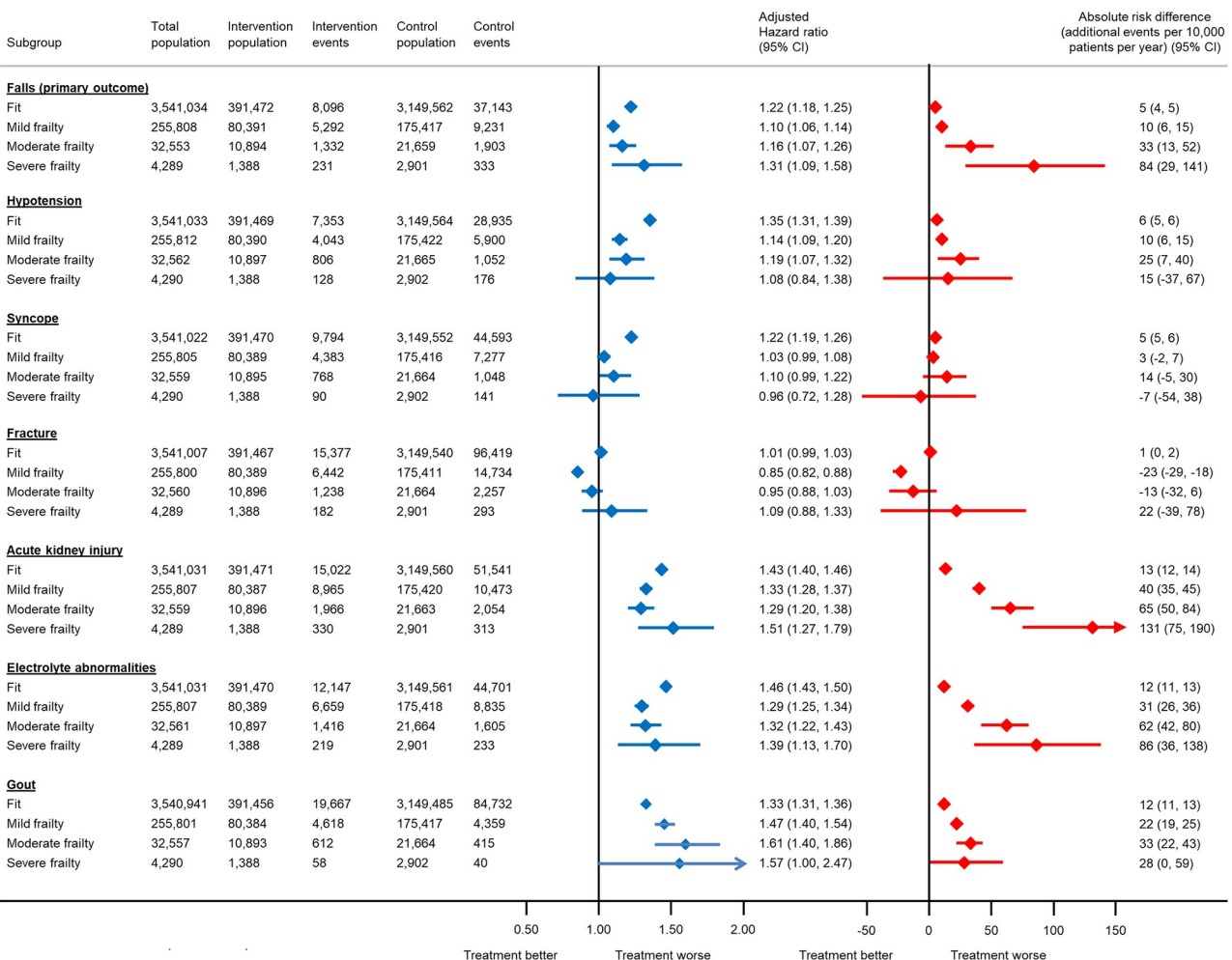

**Fig 3. Association between antihypertensive treatment and serious adverse events leading to hospitalisation or death, by frailty status at the index date.** The total number of patients included in each analysis varies due to the exclusion of participants who experienced the outcome of interest on the index date. Models adjusted for propensity score. CI, confidence interval.

these associations vary by age and frailty. Data were taken from 2 databases of electronic health records, covering more than half the population in England, and representative on the basis of age, ethnicity, and deprivation [12,13]. Outcomes were based on secondary care data on the primary cause of hospitalisation or death and, therefore, were less likely to be biased by any knowledge that patients were taking antihypertensive therapy.

Treatment effect estimates fell within the 95% CIs of estimates from meta-analyses of randomised controlled trials [9] for all outcomes except hypotension and acute kidney injury. These discrepancies may be partly explained by differences in the way outcomes were measured in this study and previous trials; hypotension is more likely to be detected in a randomised controlled trial where blood pressure is measured at regular intervals in a standardised manner. Similarly, the definition of acute kidney injury used here was based on clinical codes at hospital admission (or death) and maybe different from that used in trials where kidney function is measured much more closely.

The present analyses used an intention-to-treat approach and did not account for those patients starting treatment in the control group (936,404; [28%]). This makes the findings

comparable to randomised controlled trials but potentially underestimates the true adverse event signal. More complex analyses are possible to address this issue, for example, using time-varying covariates [30]; however, the addition of such analyses to those already employed (multiple imputation with propensity scores) would require further statistical assumptions, which are not well understood in the literature. Given that patients in the exposure group were typically exposed to treatment for much longer than those in the nonexposed group (median 0 years versus 6 years), these treatment effect estimates should be viewed as conservative estimates of the true underlying adverse event signal. Given that these data could be used to justify not starting treatment, which could carry some benefit, such conservative estimates are appropriate.

Analyses focussed on new users of antihypertensive treatment to avoid the observed associations being confounded by previous prescription of antihypertensive treatment. However, this means that the results only reflect the association between antihypertensive treatment and serious adverse events in patients within 10 years of starting therapy. The association between treatment and serious adverse events in those already prescribed therapy for longer periods may be different. Further, baseline covariates were defined at the index date, rather than cohort entry, so it is possible that some covariates reflected the characteristics of patients after treatment was prescribed, although the impact of this is likely to have been small given the short exposure period.

It is possible that those patients included in the exposure group were more likely to be truly hypertensive (hence receiving antihypertensive treatment), and therefore, the observed differences in outcomes reflect this difference in hypertensive status rather than differences in treatment prescription. However, this study focussed on adverse events associated with treatment, which are predominately independent from blood pressure (with the exception of hypotension). Hypertension itself would not necessarily increase the risk of these adverse events, and conversely getting blood pressure under control through lifestyle modification would not necessarily reduce the risk of falls, syncope, or other adverse events of interest.

In addition, the study start date was 1 January 1998, and more than 50% of the cohort entered the cohort in the last 15 years. Although antihypertensive prescribing trends may have changed during the whole study period, it is unclear whether this would affect the associations observed in the present study. These prescribing trends reflect those in the United Kingdom during the study period and may have differed from those in other parts of the world. Ideally, these results should be replicated using data from other countries with different population characteristics and different antihypertensive treatment regimens.

## Implications for clinical practice

Clinical guidelines for the management of hypertension recommend greater consideration of the benefits and harms of antihypertensive treatment as patients get older and develop frailty [3,6,7], but there are limited empirical data that support this decision-making. The present analysis demonstrates a clear association between antihypertensive treatment and serious adverse events, which, for outcomes such as falls, syncope, acute kidney injury, and electrolyte abnormalities, increases with age and moderate to severe frailty.

For many patients, it is likely that both the benefits and harms of treatment will increase as they get older. In younger patients, the present analysis showed the NNH with treatment over 5 years for serious falls was negligible, meaning that the benefits of treatment clearly outweigh any harms. By contrast, in older patients (80 to 90+ years), the $NNH_5$ for serious falls was 20 to 33 (or 24 for those with severe frailty), while the $NNT_5$ for major cardiovascular events is 38 [21]. In these older patients, the benefits and harms of treatment are much more finely balanced.

There are many other factors beyond age and frailty that will affect an individual's likelihood of benefit and harm from treatment. A better understanding of these would enable physicians and patients to make more personalised treatment decisions. This approach is common place in the context of anticoagulation for patients with atrial fibrillation [31], where prediction models are used to estimate an individual's risk of stroke (and likelihood of benefiting from treatment) [32] and weigh this against their risk of a serious bleed (which may be exacerbated by treatment) [33]. Such models are now being developed in the context of antihypertensive treatment prescription [34,35], and these are needed to facilitate personalised treatment decisions based on an individual's risk and personal preferences. It is likely to be some time before these are available in routine clinical practice, so in the meantime, the results from our study offer an important insight as to when one might want to consider intervening to prevent an individual from suffering adverse events from antihypertensive treatment. These data should therefore be used to inform future health economic modelling and support evidence-based prescribing recommendations.

## Conclusions

In previously untreated patients with raised systolic blood pressure, antihypertensive treatment was associated with an increased risk of serious adverse events. Overall, the absolute risk of this harm was low, with the exception of older patients and those with moderate to severe frailty, where the risks were similar to the likelihood of benefit from treatment. In such patients, decisions about initiating or continuing antihypertensive treatment are much more finely balanced, and this should be reflected in clinical guidelines and advice given by clinicians.

## Supporting information

**S1 Extended Methods. Extended methods.**
(DOCX)

**S1 Fig. Definition of time periods used to define the cohort and follow-up periods.** Patients were eligible at cohort entry if they were aged 40 years or older, registered at a linked, "up-to-standard" general practice, had records available after the study start date (1 January 1998), had no previous prescription of antihypertensive therapy and a single systolic blood pressure reading between 130–179 mm Hg.
(DOCX)

**S2 Fig. Flow diagram showing selection of patient records for inclusion in the study.** CPRD, Clinical Practice Research Datalink; mm Hg, millimetres of mercury.
(DOCX)

**S3 Fig. Percentage of patients with an index date in each year of the observational period.**
(DOCX)

**S4 Fig. Propensity score model performance.** AUC, area under the curve; CITL, calibration in the large; E:O, expected over observed ratio.
(DOCX)

**S1 Table. Code lists to define serious adverse event outcomes.**
(DOCX)

**S2 Table. Drug classes included in the analysis.**
(DOCX)

**S3 Table. Baseline characteristics within each dataset (CPRD Gold vs. CPRD Aurum).**
*Proportions based on the number of patients with data available (i.e., excluding those with missing values) †IMD, indices of multiple deprivation; IMD score of 5 indicates patients in the highest quintile of deprivation (most deprived).
(DOCX)

**S4 Table. Antihypertensive drug exposure during the study.** ACE, angiotensin-converting enzyme; IQR, interquartile range; SD, standard deviation. Data are based on the primary analysis examining and censoring fall events.
(DOCX)

**S5 Table. Propensity score model.** CI, confidence interval; DBP, diastolic blood pressure; HDL, high-density lipoprotein; IMD, indices of multiple deprivation; SBP, systolic blood pressure.
(DOCX)

**S6 Table. Baseline characteristics of the propensity score matched cohort.** *Proportions based on the number of patients with data available (i.e., excluding those with missing values). †IMD, indices of multiple deprivation; IMD score of 5 indicates patients in the highest quintile of deprivation (most deprived).
(DOCX)

**S7 Table. Post hoc analyses of any serious adverse event and mortality.** *Models adjusted for propensity score. ‡Number need to treat to prevent 1 event.
(DOCX)

**S8 Table. Sensitivity analyses examining assumptions about missing data and competing risks (falls outcome).** *For the model accounting for competing risks, the sub-hazard ratio is presented.
(DOCX)

**S1 RECORD Checklist. S1 Checklist.**
(DOCX)

**S1 Protocol. ISAC Protocol.**
(DOCX)

## Acknowledgments

We thank Lucy Curtin for administrative support throughout the project. We thank Richard Riley and Margaret Ogden for their contributions as STRATIFY Investigators to the project. This work uses data provided by patients and collected by the NHS as part of their care and support. We are very grateful to all those patients who permit their anonymised routine NHS data to be used for this approved research.

The views expressed are those of the author(s) and not necessarily those of the NIHR or the Department of Health and Social Care.

## Author Contributions

**Conceptualization:** James P. Sheppard, Richard Stevens, Amitava Banerjee, Andrew Clegg, F. D. Richard Hobbs, Richard J. McManus.

**Data curation:** Constantinos Koshiaris, Sarah Lay-Flurrie, Subhashisa Swain.

**Formal analysis:** James P. Sheppard, Constantinos Koshiaris.

**Funding acquisition:** James P. Sheppard, F. D. Richard Hobbs, Richard J. McManus.

**Investigation:** James P. Sheppard, Richard Stevens, Sarah Lay-Flurrie, Amitava Banerjee, Brandon K. Bellows, Andrew Clegg, F. D. Richard Hobbs, Rupert A. Payne, Juliet A. Usher-Smith, Richard J. McManus.

**Methodology:** James P. Sheppard, Constantinos Koshiaris.

**Project administration:** James P. Sheppard.

**Resources:** James P. Sheppard.

**Supervision:** Constantinos Koshiaris, Richard Stevens, Sarah Lay-Flurrie, Amitava Banerjee, Brandon K. Bellows, Andrew Clegg, F. D. Richard Hobbs, Rupert A. Payne, Juliet A. Usher-Smith, Richard J. McManus.

**Writing – original draft:** James P. Sheppard.

**Writing – review & editing:** James P. Sheppard, Constantinos Koshiaris, Richard Stevens, Sarah Lay-Flurrie, Amitava Banerjee, Brandon K. Bellows, Andrew Clegg, F. D. Richard Hobbs, Rupert A. Payne, Subhashisa Swain, Juliet A. Usher-Smith, Richard J. McManus.

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
