## [Editor Report · Decision Letter 0]

12 Nov 2022

Dear Dr Sheppard, 

Thank you for submitting your manuscript entitled "The association between antihypertensive treatment and serious adverse events by age and frailty: an observational cohort study of 3.8 million patients followed up for a decade" for consideration by PLOS Medicine.

Your manuscript has now been evaluated by the PLOS Medicine editorial staff as well as by an academic editor with relevant expertise and I am writing to let you know that we would like to send your submission out for external peer review.

Please re-submit your manuscript within two working days, i.e. by Nov 16 2022 11:59PM.

Kind regards,

Callam Davidson

Senior Editor

PLOS Medicine

---

## [Decision Letter · Decision Letter 1]

4 Jan 2023

Dear Dr. Sheppard,

Thank you very much for submitting your manuscript "The association between antihypertensive treatment and serious adverse events by age and frailty: an observational cohort study of 3.8 million patients followed up for a decade" (PMEDICINE-D-22-03557R1) for consideration at PLOS Medicine. 

[LINK]

In light of these reviews, I am afraid that we will not be able to accept the manuscript for publication in the journal in its current form, but we would like to consider a revised version that addresses the reviewers' and editors' comments. Obviously we cannot make any decision about publication until we have seen the revised manuscript and your response, and we plan to seek re-review by one or more of the reviewers. 

We expect to receive your revised manuscript by Jan 25 2023 11:59PM. Please email us (plosmedicine@plos.org) if you have any questions or concerns.

We look forward to receiving your revised manuscript. 

Sincerely,

Callam Davidson, 

PLOS Medicine

plosmedicine.org

Please revise your title to ‘The association between antihypertensive treatment and serious adverse events by age and frailty in England: a cohort study’.

Please structure your abstract using the PLOS Medicine headings (Background, Methods and Findings, Conclusions).

Abstract Background: Provide the context of why the study is important. The final sentence should clearly state the study question.

Abstract Methods and Findings:

* Please ensure that all numbers presented in the abstract are present and identical to numbers presented in the main manuscript text.

Data Availability Statement: To allow interested parties to make further enquiries regarding data access, please include contact addresses (website or email) for the CPRD and NHS Digital.

Please place citations in normal script, within square brackets, and preceding punctuation.

Please cite your Supporting Information per our guidelines: https://journals.plos.org/plosmedicine/s/supporting-information

Please confirm that the CPRD Independent Scientific Advisory Committee provided ethical approval for the study (or provide the name of the institutional review board that did). 

Please ensure that the study is reported according to the RECORD guideline, and include the completed RECORD checklist as Supporting Information. Please add the following statement, or similar, to the Methods: "This study is reported as per the REporting of studies Conducted using Observational Routinely-collected Data (RECORD) guideline (S1 Checklist)."

The RECORD guideline can be found here: https://www.record-statement.org/checklist.php

Please state whether your study had a prospective protocol or analysis plan early in the Methods section.

If a prospective analysis plan was used in designing the study, please include the relevant prospectively written document with your revised manuscript as a Supporting Information file to be published alongside your study, and cite it in the Methods section. A legend for this file should be included at the end of your manuscript. 

Changes in the analysis-- including those made in response to peer review comments-- should be identified as such in the Methods section of the paper, with rationale.

Please provide the unadjusted comparisons as well as the adjusted comparisons in Table 2.

The asterisks in the legends of Figures 1-3 lack a corresponding flag in the Figures themselves. 

Please present and organize the Discussion as follows: a short, clear summary of the article's findings; what the study adds to existing research and where and why the results may differ from previous research; strengths and limitations of the study; implications and next steps for research, clinical practice, and/or public policy; one-paragraph conclusion.

Please temper claims of primacy of results by stating, "to our knowledge" or something similar.

Please remove the ‘Conflicts of interest’, ‘Funding’, and ‘Data sharing statement’ sections from the manuscript – this information is captured via the Submission Form Questionnaire.

Please delete the ‘Role of the funder/sponsor’ section.

Comments from the reviewers:

Reviewer #1: General remarks:

This very large population-based, observational study importantly adds to current literature by detailing the probabilities of adverse effects in various age and frailty groups. Because RCTs usually include better-functioning patients, the results are apt to fill gaps in current knowledge. I do not have relevant criticisms about methods and analyses, which seem appropriate for the purpose. My main comments are for the discussion and interpretation of results.

1. When considering treatment of older patients, a fundamental question is whether the treatment is ongoing or started when they enter old age. The results promote caution when treating older and frail people with antihypertensives. However, as far as I understand, participants were included without previous treatment. Therefore, the results may give a wrong signal about treatment in older people who are already using the drugs when entering old age. Cardiovascular risk including hypertension is a risk factor of frailty, so the fact that hypertension treatment should be started for risk patients before old age and before they get frail. Deprescribing due to old age alone may also involve risks. 

Please discuss those points, eg. in Limitations.

2.Line 65 " Antihypertensives are also the most commonly prescribed medications in patients admitted to hospital with adverse drug reactions". My comment: Maybe "AMONG the most commonly prescribed medications..." would be more appropriate. Often anticoagulants/NSAIDs/insulin/aspirin are holding the first place?

3. The extensive and useful review (Benetos A, et al.Circ Res. 2019 Mar 29;124(7):1045-1060 could be added to reference list, especially as it calls out for studies like yours.

4. In RCTs like HYVET and SPRINT the authors have also considered frailty in relation to their results (Warwick J, et al. BMC Med. 2015 Apr 9;13:78). I agree that those trials did not include most frail patients, nevertheless, participants were old. I think it would be appropriate to acknowledge those substudies and their message

Reviewer #2: 

This paper addresses a very important topic that can, as investigators state, help inform clinical decision making about net benefits and harms of antihypertensive treatment. This is a particularly relevant issue given the recent move towards more aggressive blood pressure targets. The manuscript is well written and there are strengths in the design and population as I note. Because of the potential importance of results, I think it is important for investigators to address some important issues.

Design The clinical population-based observational study is probably the optimal approach to determining adverse effects of antihypertensive treatment as clinical trial data has the inherent limitations of selection bias as well as relatively short follow up. Replicating the beneficial outcomes associated with antihypertensive treatment seen in clinical trials further enhances confidence in this observational study despite the inherent limitations of observational data. 

The study start date (1st January 1998) means that these data are now up 25 years old. Antihypertensive treatment regimens have changed; adverse effects of current antihypertensive regimens may differ from those studied. For example, investigators note that, "(63.6%) patients were prescribed one antihypertensive medication, 131,342 (27.1%) were prescribed two and 45,139 (9.3%) were prescribed three or more medications." at least in the United states, the prevalence of two or three antihypertensives are much higher than in this study. Furthermore, the classes of antihypertensive medications have likely changed. For example, a much smaller percentage of individuals in this court were on calcium channel blockers than today. this should at a minimum be addressed in the discussion section. Preferably, there would be a secondary analysis, limited to more recent data.

Population 

- A strength of this study is the large and representative sample, although limited to a single country. Results will need to be replicated in other countries with different population characteristics and different antihypertensive treatment regimens.

-The focus of the paper is on adverse events from antihypertensives. The population of significant clinical concern in which it is less clear whether the benefits outweigh the harms is the older population. There is little doubt that benefits outweigh harms in younger populations. Therefore, why was the population 40 and older rather than the clinically relevant population such as 60 or 65 and older? Similar to limiting systolic blood pressure to those <180, this study would be strengthened by limiting the population to those in whom there is question benefit outweighing harm. While secondary data by age are presented in the appendix, the primary results are diluted by the younger population.

- Of note, a surprisingly small percentage of individuals with hypertension seem to have received antihypertensive treatment anytime during the 10 years of follow up,. It appears that only 12% were on treatment at baseline and only 28% of those not on treatment initially received treatment during follow up, meaning only 40% were ever treated. This is a much smaller number than would be expected in the United States for example. Perhaps this may be related to the fact that they define hypertension as greater than 130 systolic. Treatment cut off of 140 systolic would have been clinical indicator for treatment during the time period of this study. therefore, systolic 140-180 appropriate range for this study

Definition of antihypertensive treatment

"Exposure to antihypertensive medication was defined by the most recent prescriptions in the 12 months following cohort entry. The index date was defined at the end of this exposure period, after which patients were followed up for up to ten years. " Patients were defined as on or off treatment based on treatment status of the index date. obviously, there are going to be many changes in treatment over 10 years of follow up. Assuming treatment versus non treatment over 10 years based on one point in time, while perhaps matching "intention to treat" approach of randomized controlled trials does not provide confidence that adverse effects can be linked to antihypertensive treatment as participants may or may not been on treatment at the time. Analysis according to antihypertensive treatment exposure byintervals such as month would seem to be a better approach to the stated aims of this study. Investigators appear to have had access to antihypertensive prescriptions over the 10 years of follow up based on eTable 4 data which showed that 28% of "non exposed" participants received anti hypertensive during follow up. Investigators state in the discussion that their approach provides a conservative estimate of risk of adverse effects with antihypertensive treatment. They further note that more complicated analysis would be subject to other biases. While this is true, finding similar relationship between antihypertensive treatment and adverse outcomes with this additional analysis would provide greater confidence in the results.

Outcome

- The individual adverse effects included seem appropriate. However, they were each studied as separate outcomes. To better evaluate the overall adverse effects of antihypertensive treatment, a composite outcome of the serious adverse effects would be most appropriate from a clinical perspective. This would parallel the use of MACE cardiovascular outcomes. And many studies, a single outcome such as stroke does not reach statistical significant while the composite outcome does.

- As the relevant clinical question is net benefit versus harm of antihypertensives, it would be helpful, at least as a secondary analysis, to see the relationship between antihypertensive treatment and all cause mortality, the relevant survival outcome for older adults, particularly those who are frail, who have multiple contributors to mortality.

Summary: This paper addresses a very important topic and can, as investigators state, help inform clinical decision making. Given its importance, I would urge the investigators to do the analysis and present the data that can strengthen confidence and results. I would suggest limiting the analysis to the older population with systolic BP 140-180 for whom the question of benefit versus harm of antihypertensives is clinically relevant. Second, I would suggest doing the time varying analysis to see that the relationship between exposure and adverse effects holds up. Third, I think it would be helpful to have a composite measure of adverse effects to parallel MACE. 

Reviewer #3: Alex McConnachie, Statistical Review

Sheppard and colleagues present the results of a very large retrospective cohort study, looking at the association between the use of antihypertensive medications, and the incidence of a variety of adverse outcomes. This review considers the statistical aspects of the paper.

Overall, these are very good. The definition of the cohort, exposure, and outcomes, are very clear. Various methods of propensity score and covariate adjustments are applied. Cox proportional hazards models are used to assess associations, and a competing risks analysis is done as a sensitivity analysis. The results are generally presented very clearly. Both relative and absolute differences are reported. Whilst the paper recognizes that more complex analyses could have been done, I agree that this would be too much - there is plenty in the paper as it stands, and the "ITT-like" approach is very robust.

I do have quite a few comments, though I believe many are to do with the way things are described or presented.

The main thing that confused me was the derivation of the propensity scores. Line 130 states that "Propensity scores were generated for each outcome of interest…" It is not clear to me why multiple propensity scores are being used. To me, the propensity score is based on the predicted probability of being exposed, and for each outcome, the exposure is the same. Surely the propensity score has nothing to do with the outcomes?

However, lines 177 to 180 make it sounds more like I would expect - talking about predicting the likelihood of treatment, though in the supplement, eTable 5 is titled "Propensity score model (primary outcome [falls])" - I do not see why the PS model relates to the primary outcome in particular. eTable 6 then reports O/E ratios and AUCs in relation to each outcome, even though the PS model is predicting treatment. And, if the PS models are specific to each outcome, it seems odd that the O/E ratios and AUCs should be virtually identical for all outcomes. Similarly, the panels of eFigure 3 are indistinguishable, to my eye, so it is not clear to me what they are showing.

I note that in eTable 5, the continuous variables (e.g. age, BMI, BP, Cholesterol) are categorised for PS modelling. Would it be better to treat these as continuous predictors?

The index date is taken as 12 months after the first SBP recording of 130 or more, and the exposure as being on antihypertensive medication at the index date. The unexposed group are described in the Tables as "No antihypertensive prescription", but this is not entirely accurate, I think. Exposure requires a prescription for at least 30 days, within the last 30 days prior to the index date. The unexposed could then include people who took some medication between cohort entry and the index date. It might be good to show some summary statistics in this regard - how many people were there like this? How much medication did they take?

The covariates for adjustment and derivation of propensity scores are defined using data up to the index date. Some of this data could therefore be after the point at which someone was put onto antihypertensive medication, which could happen at any time between cohort entry and the index date. Would it be safer to use data prior to cohort entry to define the covariates?

One interpretation of the results could be that whether or not someone ends up in the exposed group is simply a measure of whether the initial SBP reading of 130 or more was a true or a false positive. For many of those in the unexposed group, that first raised blood pressure might have been an isolated result. In comparison, those meeting the definition of exposure might represent a truly hypertensive group. Alternatively, some in the unexposed group might have been stimulated by their initial SBP reading to adopt a healthier lifestyle, and got their BP back under control without the need for medication, whereas the exposed group may have been more resistant to such efforts, so that the unexposed group are simply a healthier group in general. Even if the covariate and PS adjustment can account for these differences between the two groups, the differences in outcomes are not necessarily due to the exposure, since exposure (as defined in this study) could (at least in part) be a marker of true hypertension, or for the severity of whatever is causing the hypertension. I think the limitations section should say more about alternative explanations for the results.

There is no mention of the proportionality of hazards in the Cox models. How was this checked, and was the assumption OK?

The subgroup analyses are a good feature of the paper, showing how the risk/benefit balance shifts according to the underlying risk of the outcomes. The results state that there were no differences in relative associations between exposure and outcomes across subgroups, but looking at Figures 2 and 3, I am not totally convinced. Having some interaction p-values might help, but for me it definitely looks like there might be some trends in relative associations across age groups, in particular. The data must be available to test for interactions, either using categories of age, or with age as a continuous modifier of the associations.

[LINK]

---

## [Decision Letter · Decision Letter 2]

16 Feb 2023

Dear Dr. Sheppard,

Thank you very much for re-submitting your manuscript "The association between antihypertensive treatment and serious adverse events by age and frailty: a cohort study" (PMEDICINE-D-22-03557R2) for review by PLOS Medicine.

I have discussed the paper with my colleagues and the academic editor and it was also seen again by one reviewer. I am pleased to say that provided the remaining editorial and production issues are dealt with we are planning to accept the paper for publication in the journal.

[LINK]

We look forward to receiving the revised manuscript by Feb 23 2023 11:59PM.   

Sincerely,

Callam Davidson, 

Senior Editor 

PLOS Medicine

plosmedicine.org

Requests from Editors:

Please label hazard ratios as adjusted hazard ratios (aHR) throughout the manuscript, as applicable. 

Lines 92-94: Please update this bullet point in the author summary for clarity – I feel that the current wording could be confusing for a non-scientific audience, particularly number needed to treat.

What does the * in the S2 Table signify?

Line 223: ‘are detailed’.

S7 Figure: X-axis is truncated.

Please check your Supporting Information for instances of the word ‘gender’ where ‘sex’ would be more appropriate. 

Please remove any italics formatting from the References.

Please check Reference 15 for missing information.

Comments from Reviewers:

Reviewer #3: Alex McConnachie, Statistical Review

I thank the authors for their consideration of my original points. I would have liked to have seen p-values for tests of differences in associations between subgroups, but it is good that they have added some more commentary about this. Otherwise I am happy with the authors' responses to my original points, and I have no further comments to make.

[LINK]

---

## [Editor Report · Decision Letter 3]

20 Mar 2023

Dear Dr. Sheppard,

Thank you very much for re-submitting your manuscript "The association between antihypertensive treatment and serious adverse events by age and frailty: a cohort study" (PMEDICINE-D-22-03557R3) for review by PLOS Medicine.

I have discussed the paper with my colleagues and the academic editor. I am pleased to say that provided the remaining editorial and production issues are dealt with we are planning to accept the paper for publication in the journal.

The remaining issues that need to be addressed are listed at the end of this email. 

We look forward to receiving the revised manuscript by .   

Sincerely,

Callam Davidson, 

Associate Editor 

PLOS Medicine

plosmedicine.org

Requests from Editors:

Abstract -- methods: Please briefly state which variables enter the propensity score.

Abstract – methods and findings: Before the sentence “Although treatment effect estimates fell within the 95% CIs of those from randomised trials, these analyses were observational in nature and so bias from unmeasured confounding cannot be ruled out.” it would be good to strongly emphasize that this data is superior to data from trials in that it provides a view of real-life effectiveness – the trial is unlikely to be externally valid, because trial participants are typically highly selected and diligently supported by the trial team in their treatment retention and adherence. 

It would be good to also emphasize this strength of the present study in the “Why was this study done?” section in the Author Summary.

Abstract – methods and findings: Finally, it would also be good to mention that the results remained essentially the same when the competing risk of death from all causes was taken into account in sensitivity analyses.

---

## [Editor Report · Decision Letter 4]

24 Mar 2023

Dear Dr Sheppard, 

On behalf of my colleagues and the Academic Editor, Dr Sanjay Basu, I am pleased to inform you that we have agreed to publish your manuscript "The association between antihypertensive treatment and serious adverse events by age and frailty: a cohort study" (PMEDICINE-D-22-03557R4) in PLOS Medicine.

PRESS

Sincerely, 

Callam Davidson 

Associate Editor 

PLOS Medicine